# Hierarchical Meta Reinforcement Learning for Multi-Task Environments

## Abstract

Deep reinforcement learning algorithms aim to achieve human-level intelligence by solving practical decisions-making problems, which are often composed of multiple sub-tasks. Complex and subtle relationships between sub-tasks make traditional methods hard to give a promising solution. We implement a first-person shooting environment with random spatial structures to illustrate a typical representative of this kind. A desirable agent should be capable of balancing between different sub-tasks: navigation to find enemies and shooting to kill them. To address the problem brought by the environment, we propose a Meta Soft Hierarchical reinforcement learning framework (MeSH), in which each low-level sub-policy focuses on a specific sub-task respectively and high-level policy automatically learns to utilize low-level sub-policies through meta-gradients. The proposed framework is able to disentangle multiple sub-tasks and discover proper low-level policies under different situations. The effectiveness and efficiency of the framework are shown by a series of comparison experiments. Both environment and algorithm code will be provided for open source to encourage further research.

## 1 Introduction

With great breakthrough of deep reinforcement learning (DRL) methods (Mnih et al., 2015; Silver et al., 2016; Mnih et al., 2016; Schulman et al., 2015; Lillicrap et al., 2015), it is an urgent need to use DRL methods to solve more complex decision-making problems. The practical problem in real world is often a subtle combination of multiple sub-tasks, which may happen simultaneously and hard to disentangle by time series. For instance, in StarCraft games (Pang et al., 2019), agents need to consider building units and organizing battles, sub-tasks may change rapidly over the whole game process; sweeping robots tradeoff between navigating and collecting garbage; shooting agents should move to appropriate positions and launch attacks, etc. The relationship between sub-tasks is complex and subtle. Sometimes they compete with each other and need to focus on one task to gain key advantages; at other times, they need to cooperate with each other to maintain the possibility of global exploration. It is often time consuming and ineffective to learn simply by collecting experience and rewarding multiple objectives for different sub-tasks.

A reasonable idea is to utilize deep hierarchical reinforcement learning (DHRL) methods (Vezhnevets et al., 2017; Igl et al., 2020), where the whole system is divided into a high-level agent and several low-level agents. Low-level agents learn sub-policies, which select atomic actions for corresponding sub-tasks. The high-level agent is responsible for a meta task in the abstract logic or coarser time granularity, guiding low-level agents by giving a goal, or directly selecting among sub-policies. However, DHRL methods face some inherent problems: due to the complex interaction between multi-level agents, there is no theoretical guarantee of convergence, and it shows unstable experimental performance. Most DHRL methods require heavy manual design, and end-to-end system lacks reasonable semantic interpretation. In addition, these agents are often constrained by specific tasks and are easy to overfit. Even transferring between similar tasks, they perform poorly and need a lot of additional adjustments.

We introduce a first-person shooting (FPS) environment with random spatial structures. The game contains a 3D scene from human perspective. When the player defeats all enemies, the player wins the game. When the player drops to the ground or losses all health points, the player loses the

game. It is very risky for the player to drop to the ground, thus environment contains two key tasks: navigation and combat. The terrain and enemies in the game are randomly generated. This ensures: 1) the agent cannot learn useful information by memorizing coordinates; 2) the possibility of over-fitting is restrained and the generalization ability of learned policy is enhanced. The state information is expressed in the way of raycast. This representation of environment information requires much less computing resources than the raw image representation. It can be trained and tested even with only CPU machines, which makes us pay more attention to the reinforcement learning algorithm itself rather than the computing ablity related to image processing.

For this environment, we propose a Meta Soft Hierarchical reinforcement learning framework (MeSH). The high-level policy is a differentiable meta parameter generator, and the low-level policy contains several sub-policies, which are in the same form and differentiated automatically in the training procedure. The high-level policy selects and combines sub-policies through the meta parameter and interacts with the environment. We find that the meta generator can adaptively combines sub-policies with the process of the task, and have strong interpretability in semantics. Compared with a series of baselines, the agent has achieved excellent performance in FPS environment.

The main contributions of this work are as follows:

- clarifying the complex relationship between multi-task composition.
- a novel meta soft hierarchical reinforcement learning framework, MeSH, which uses differentiable meta generator to adaptively select sub-policies and shows strong interpretability.
- a series of comparison experiments to show the effectiveness of the framework.
- an open-sourced environment and code to encourage further research on multi-task RL [1].

In this paper, we discuss the related work in Section 2. We introduce the details of the implemented environment in Section 3. We show our proposed framework in Section 4. We demonstrate details of our experiments in Section 5. At last, we conclude in Section 6.

## 2 RELATED WORK

In decision-making problems with high-dimensional continuous state space, the agent often needs to complete tasks that contain multiple sub-tasks. To complete taxi agent problem (Dieterich, 2000), the agent needs to complete sub-tasks such as pickup, navigate, putdown. Menashe & Stone (2018) proposed Escape Room Domain, which is a testbed for HRL. The agent leaves the room from the starting point and needs to press four buttons of different colors to leave the room. In these environments, the agent needs to optimize several sub-tasks and minimize the mutual negative influence between them. However, sub-tasks in these environments are timing dependent. The proposed methods above are helpless in a multi-task environment that needs to fulfill multiple tasks simultaneously.

Architectural solutions use hierarchical structure to decompose tasks into action primitives. Sutton et al. (1999) models temporal abstraction as an option on top of extended actions, Bacon et al. (2017) proposes an actor-critic option method based on it. Henderson et al. (2017) extend the options framework to learn joint reward-policy options. Besides, Jiang et al. (2019) construct a compositional structure with languages as abstraction or instruction. Due to specific structure design of these methods, high-level agent is unable to execute multiple sub-policies simultaneously in any form.

Recent HRL works learn intra-goals to instruct sub-policies. Vezhnevets et al. (2017) proposes a manager-worker model, manager abstracts goals and instructs worker. This architecture uses directional goal rather than absolute update goal. Oh et al. (2017) learns a meta controller to instruct implementation and update of sub-tasks. Igl et al. (2020) presents a new soft hierarchy method based on option, it learns with shared prior and hierarchical posterior policies. Yu et al. (2020) proposes a method that projects the conflict gradient onto the normal plane to avoid some task gradients dominating others. Compared with the hard hierarchy methods, these methods use the state's natural features to update the upper-level policy, avoiding the timing constraints of handcrafted sub-tasks. Due to the lack of meaningful learning goals of sub-policies, the low-level policies fail to focus on explainable sub-tasks.

---

[1]https://github.com/MeSH-ICLR/MEtaSoftHierarchy.git

HRL has been recently combined with meta-learning. Frans et al proposes Meta-Learning Shared Hierarchies (MLSH) (Frans et al., 2017), which updates master policy and sub-policy by meta-learning. Zou et al. (2019) uses meta-learning to learn reward shaping. Rakelly et al. (2019) extends the idea of disentangling task inference and control to help agents leverage knowledges between tasks. Although these methods reduced the limitation of policy structures, it is difficult to learn multi-tasks in parallel due to the fixed time steps selected by the master policy for every sub-policy.

It is also reasonable to learn sub-tasks based on decomposition of rewards. Saxe et al. (2017) expresses behaviors as different weighted task blends by the communication between layers so that can maintain a parallel distributed representation of tasks, but the structure is hard to abstract the relationship between different sub-tasks. Grimm & Singh (2019) and Lin et al. (2019) allocate each state with one sub-reward function, which is a non-overlapping decomposition of spatial space. In contrast to these approaches, our method can represent complex and subtle relationship between multiple sub-tasks and perform them simultaneously with explainable sub-goals.

## 3 ENVIRONMENT

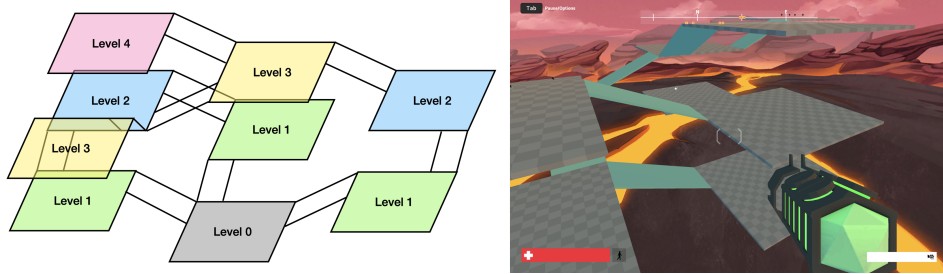

Figure 1: First-person shooting environment with random terrain.

To better understand multi-task decision-making problems, we firstly introduce a first-person shooting environment with random terrain, as shown in Figure 1. The game contains a 3D scene, which analogy to human perspective. This makes the behavior of trained agent similar to human intelligence. The real-time information is shown in Table 1. The condition of winning the game is to defeat all the enemies in the game. When the player drops to the ground or losses all health points, the game will be judged as failed. This game is very risky as it's easy to drop. Thus the environment contains two key tasks: navigation and combat. The terrain and enemies in the game are randomly generated. This ensures: 1) the agent cannot learn useful information by memorizing coordinates; 2) the possibility of over-fitting is restrained and the generalization ability of learned policy is enhanced. The state information of the agent is expressed in the way of raycast, as shown in Figure 2. This representation of environment information requires much less computing resources than raw image representation. It can be trained and tested even with only CPU machines, which makes us pay more attention to the reinforcement learning algorithm itself rather than the computing ability related to image processing.

The generation rules of random terrain are as follows. The maximum generated height of random terrain is set to 5. In the initial state, there is only one parcel, which is also the place where the player is born. We add this parcel to the parcel queue. If the parcel queue is not empty, the parcel at the head of the queue will be taken out. When the maximum height is not reached, we expand the parcel to four directions of the higher level with equal probability. If there is no new parcel in the corresponding position of the higher level, a new parcel is generated. A ramp is established between the new parcel and the current parcel and a random number of enemy is created at the random position of the new parcel, and then the new parcel is added to the parcel queue. If there are new parcels in the corresponding position of the higher level, the new parcels are not generated repeatedly, and only the ramp between the two parcels is added. Repeat adding parcels until the parcel queue is empty, then terrain generation is completed.

The FPS environment is typically a combination of two sub-tasks: navigation and combat. The relationship between these two sub-tasks is subtle and complex. Sometimes they compete with

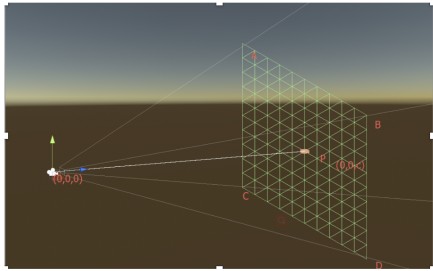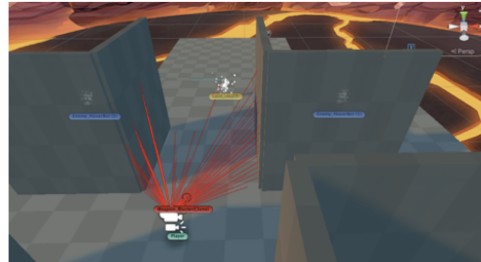

Figure 2: The raycast information. The range of the game camera is 60 degrees of vertical view, 90 degrees of horizontal view, and the interval of each ray is 5 degrees. Therefore, a total of 247 = (60 / 5 + 1) * (90 / 5 + 1) rays are emitted. The raycast return {object, distance, object, distance, ⋯} as a sequence. The object is represented by {0: none, 1: player itself, 2: mesh, 3: blood bag, 4: enemy}, and the distance representation is direct transmission value. When there is no object touched, the distance is also 0.

Table 1: Real-time Information

| Field Name | Field Type | Simple Description | Data Range |
|---|---|---|---|
| hp | float | player's health point | [0, 100] |
| energy | float | player's firing energy | [0, 16] |
| x | float | player's x-coordinate | [-30, 20] |
| y | float | player's y-coordinate | [-1, 16] |
| z | float | player's z-coordinate | [-80, 10] |
| rotation_x | float | x-coordinate of player's orientation | [0, 360] |
| rotation_y | float | y-coordinate of player's orientation | [0, 360] |
| rotation_z | float | z-coordinate of player's orientation | [0, 0] |
| raycast | repeated int32 | player's vision for object & distance | object: 0-4 distance: $[0, +\infty]$ |
| kill | int32 | number of enemies killed | $[0, +\infty]$ |

each other, but at other times, they cooperate for a common objective. For instance, in navigation missions, in order to explore more unseen terrain, sometimes we need to fight to clear enemies along the way; but at other times, we have to focus on navigation to pass through narrow terrain. Similarly, in combat missions, sometimes we need to move to get a better position to shoot; but at other times, we need to focus on shooting to kill the enemy quickly. This environment is very representative of practical problems, since a large number of them can be divided into several parts which are contradictory and unified. In addition, each sub-task in this environment is simple and clear, but the combination of them greatly increases the difficulty of solving the problem. This forces the RL algorithm to focus more on dealing with these complex relationships, rather than the specific techniques for solving a single problem.

## 4 POLICY OPTIMIZATION

### 4.1 FRAMEWORK

The proposed MeSH framework includes two policies: high-level policy and low-level policy, as shown in Figure 3. The high-level policy is a differentiable meta parameter generator, and the low-level policy contains $N$ sub-policies, which are in the same form and correspond to $N$ sub-tasks respectively. The high-level policy automatically selects and combines sub-policies of the low-level through the meta parameter generator and interacts with the environment.

In the proposed framework, firstly, shared encoder layer and RNN layer are deployed to learn the environmental state representation $s_t$ from the observation history. Based on $s_t$, a high-level meta-

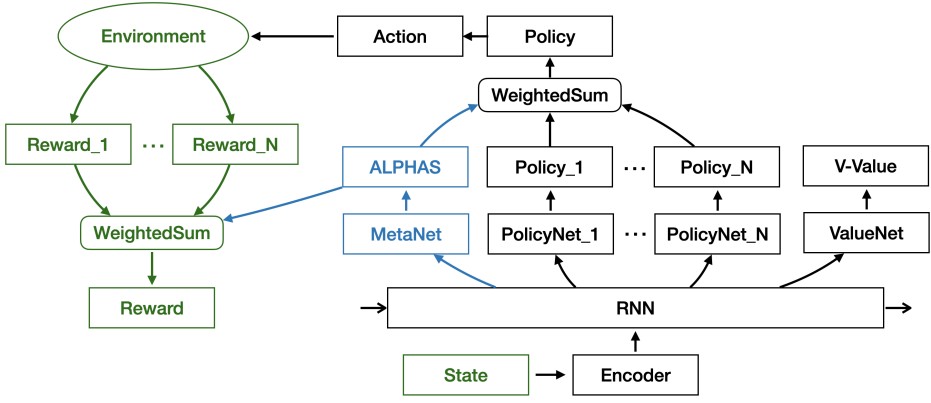

Figure 3: The MeSH framework.

parameter network is established to generate meta-parameters $\alpha = (\alpha_1, \alpha_2, \cdots, \alpha_N)$. And $N$ low-level policy networks are established to generate $N$ different sub-policies $(\pi_1, \pi_2, \cdots, \pi_N)$, respectively. The policy $\pi$ ultimately used to choose actions is the weighted sum of the $N$ policies:

$$\pi = \sum_{i=1}^{N} \alpha_i \cdot \pi_i. \tag{1}$$

When the policy $\pi$ chooses the action and interacts with the environment, the environment moves to next state and returns corresponding rewards $(R_1, R_2, \cdots, R_N)$. Among them, $R_i$ is the reward for corresponding sub-tasks. The final reward $R$ received by the agent is weighted sum of the $N$ rewards:

$$R = \sum_{i=1}^{N} \alpha_i \cdot R_i. \tag{2}$$

This setting has two advantages: 1) $\alpha$ can automatically select the weight of the corresponding policy according to the state of the environment; 2) derive the differentiation of the $N$ policies in an implicit way.

In the training process, we use IMPALA (Espeholt et al., 2018) as the basic framework of large-scale distributed reinforcement learning. Since the process of sample collection and parameter updating are decoupled, learning is off-policy and V-trace technique is used to reduce this difference. The *Loss* of the framework is

$$Loss = c_1 \cdot \min(clip(\rho) \cdot A, \rho \cdot A) + c_2 \cdot MSE(v, vs) + c_3 \cdot Entropy. \tag{3}$$

where $A$ and $vs$ are $advantages$ and target $V$-value estimated by V-trace.

Due to the complexity of composite tasks, it is usually difficult to obtain positive reward from naive policy. We utilize the idea of self-imitation learning (SIL) (Oh et al., 2018) to speed up the learning of positive behaviors. Specifically, the original SIL algorithm has not been adopted. Only those samples whose return exceeds the current value estimation are saved in a special buffer, from which a mini-batch data is extracted and learned together with the normal samples in every update step.

## 4.2 META-GRADIENT

We divide the extracted buffer into two parts to calculate the loss to be optimized respectively, which is denoted as $L_{train}$ and $L_{val}$. Both losses are determined not only by meta-parameter $\alpha$ but also the parameters of policy networks $\omega$. This implies a bilevel optimization problem with $\alpha$ as the upper-level variable and $\omega$ as the lower-level variable (Liu et al., 2018):

$$\min_{\alpha} \quad L_{val}(\omega^*(\alpha), \alpha),$$
$$\text{s.t.} \quad \omega^*(\alpha) = \arg\min_{\omega} \ L_{train}(\omega, \alpha). \tag{4}$$

Due to the expensive inner optimization of evaluating the gradient exactly, we use the approximation scheme as follows:

$$\bigtriangledown_\alpha L_{val}(\omega^*(\alpha), \alpha) \approx \bigtriangledown_\alpha L_{val}(\omega - \xi \bigtriangledown_\omega L_{train}(\omega, \alpha), \alpha). \tag{5}$$

where $\omega$ denotes the current policy weights, and $\xi$ is the learning rate for a step of inner optimization. The idea is to approximate $\omega^*(\alpha)$ by adapting $\omega$ using only a single training step, without solving the inner optimization completely. Denote $\hat{\omega} = \omega - \xi \bigtriangledown_\omega L_{train}(\omega, \alpha)$, we can approximate (5) by

$$\begin{aligned}
&\bigtriangledown_\alpha L_{val}(\omega^*(\alpha), \alpha) \\
\approx &\bigtriangledown_\omega L_{val}(\hat{\omega}, \alpha) \cdot (-\xi \bigtriangledown_{\omega,\alpha} L_{train}(\omega, \alpha)) + \bigtriangledown_\alpha L_{val}(\hat{\omega}, \alpha) \\
\approx &\bigtriangledown_\omega L_{val}(\hat{\omega}, \alpha) \cdot -\frac{\xi}{2\epsilon} \cdot (\bigtriangledown_\alpha L_{train}(\omega + \epsilon, \alpha) - \bigtriangledown_\alpha L_{train}(\omega + \epsilon, \alpha)) + \bigtriangledown_\alpha L_{val}(\hat{\omega}, \alpha).
\end{aligned} \tag{6}$$

### 4.3 TRAINING ALGORITHM

Our hierarchical framework is end-to-end, and the influence of high-level on low-level is realized by differentiable meta parameters. Therefore, in the process of forward inference, we regard the whole framework as a whole, and no longer emphasize the concept of hierarchy. Only in the process of backward update, we need to use meta-gradient to update the meta parameters $\alpha$. Thus we will distinguish the different levels of the framework.

---

**Algorithm 1** Meta Soft Hierarchical reinforcement learning framework (MeSH).

---

1: Initialize parameters $\omega$ and $\alpha$.
2: Initialize replay buffer $D_N$ and SIL buffer $D_S$.
3: Initialize $t \leftarrow 0$
4: **while** $True$ **do**
5:     //Stage 1. Transition Generating Stage.
6:     Sample $A_t \sim \pi(A_t|S_t, \omega, \alpha)$.
7:     Generate $S_{t+1}, R_{1t}, ..., R_{Nt} \sim p(S_{t+1}, R_{1t}, ..., R_{Nt}|S_t, A_t)$.
8:     Calculate $R_t$ by (2).
9:     Store $(S_t, A_t, R_t, S_{t+1})$ in $D_N$.
10:     //Stage 2. Parameter updating stage.
11:     Sample mini-batch of transitions from $D_N$ and $D_S$.
12:     Update $\omega$ by minimizing (3).
13:     Compute and accumulate meta-gradient of $\alpha$ by (6).
14:     Update SIL buffer $D_S$.
15:     **if** $t \equiv 0 (mod\ c)$ **then**
16:         Apply meta-gradient of $\alpha$.
17:     **end if**
18:     $t \leftarrow t + 1$
19: **end while**

---

## 5 EVALUATION

In this section, we conduct a series experiments in the proposed first-person shooting environment. There are two questions we mainly focus on: 1) how the proposed framework performs compared to representative baselines; and 2) whether the proposed framework can learn different meaningful sub-policies and combine them appropriately.

### 5.1 EXPERIMENTAL SETUP

In the training process, we use IMPALA as the basic framework of large-scale distributed reinforcement learning. Four CPU-only machines are used as workers, responsible for interacting with the game environment and collecting the transition sequences. A machine with GPU serves as a learner, receives the transition sequences transmitted by the worker and updates the parameters. Since the proposed first-person shooter environment is typically a combination of two sub-tasks (navigation

and combat), we set the number of sub-policies as $N = 2$. The rewards for navigation $R_1$ are set as 0.01 per step on the level, and 2.0 per meter on the ramps. The reward for combat $R_2$ is set as 10.0 per enemy killed. Thus we hope that $\pi_1$ to learn the navigation sub-policy and $\pi_2$ to learn the combat sub-policy. We set the discount factor $\gamma = 0.997$. We use the Adam optimizer to minimize the losses, and the initial learning rate is set to $10^{-3}$ with linear decay. The time interval for meta-gradient update is set as $c = 8$. The batch size for normal buffer and SIL buffer is 512 and 64 respectively with the sequences' length set as 40. All experiments in this work use the same state encoder layer and RNN layer with LSTM units. To ensure the stability of training LSTM in the dynamic environment, we utilize the previous hidden state as initial state as introduced in R2D2 (Kapturowski et al., 2018). In addition, SIL buffer is also implemented in all experiments to accelerate the learning speed of behaviors with sparse but large reward.

## 5.2 PERFORMANCE

### 5.2.1 BASELINES

To verify the effectiveness of our proposed framework, two types of baselines are chosen for comparison. The first is the classical methods in reinforcement learning (a '+' indicates that a SIL module is added to the original method) and the second is the variant of the framework to show its ablation performance. To construct a fair comparison, we also do parameter-tuning for all baselines.

- **IMPALA+**: A typical distributed reinforcement learning framework with high throughput, scalability, and data-efficiency.
- **FuNs+**: A goal-based hierarchical reinforcement learning framework with abstract goal generated by the high-level agent and guiding the behaviors of low-level agent.
- **HardHrl**: A variant of our proposed framework with meta-parameter $\alpha$ constrained as the form of one-hot encoding, only one sub-policy is executed per step.

### 5.2.2 PERFORMANCE COMPARISON

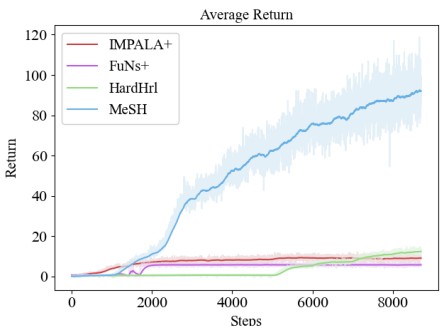

Figure 4: Average Return in Training.        Figure 5: Maximum Return in Training.

Figure 4 and Figure 5 show the average and maximum return of the collected buffer in the training process. Only the proposed framework MeSH achieves good performance, which can acquire high return in both navigation and combat sub-tasks. Other methods perform poorly. Among them, HardHrl performs slightly better than IMPALA+. From the observation of rollout result, HardHrl can execute both navigation and combat sub-tasks in a lower-level. While IMPALA+ can only execute navigation task with none of enemy killed, which shows that a single policy cannot handle multiple sub-tasks at the same time. FuNs+ has hardly learned any reasonable behavior, which indicates that abstract goal can hardly deal with the complex relationship between multiple sub-tasks simultaneously.

## 5.3 DISCUSSION ON DEALING WITH COMPLEX RELATIONSHIP

Figure 6 shows a fragment in an episode in the test process. We can observe that when the agent execute navigation sub-task, the value of $\alpha_2$ is small, which indicates that the current behavior is

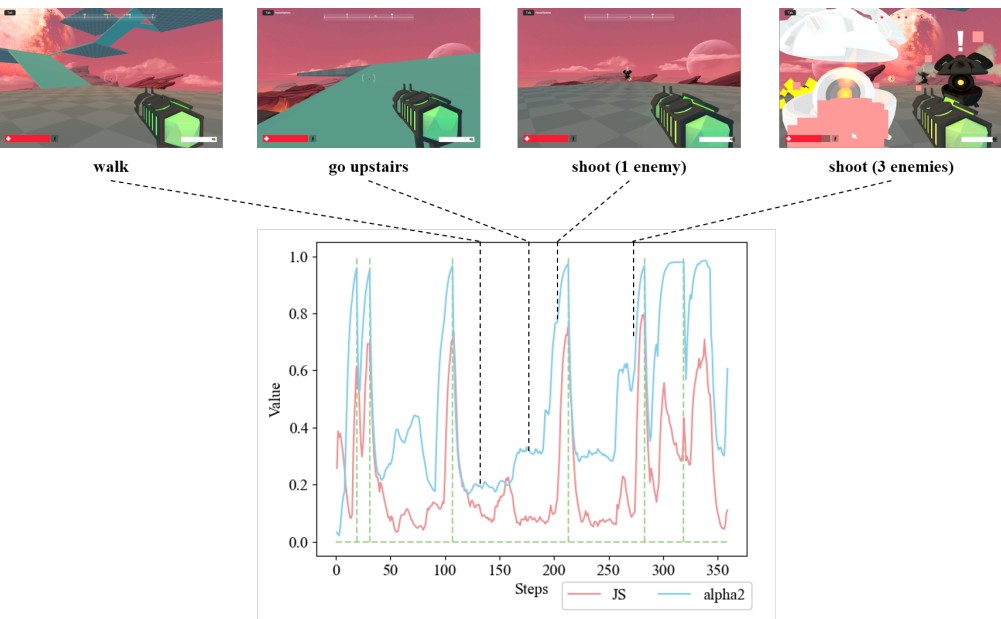

**walk**    **go upstairs**    **shoot (1 enemy)**    **shoot (3 enemies)**

Figure 6: Test Procedure. The green dashed line indicates the time step when an enemy is killed. We record $\alpha_2$ (solid blue line, the second component of meta-parameter $\alpha$, representing the weight of the combat sub-policy $\pi_2$) and JS divergence between the two learned sub-policies (solid red line) with the test process.

more influenced by the navigation policy $\pi_1$; when the agent execute combat sub-task, the value of $\alpha_2$ increases rapidly and approaches 1.0, which indicates that the current behavior is almost controlled by the combat policy $\pi_2$. Therefore, $\alpha$ can combine different sub-policies appropriately to adapt to complex conditions.

The JS divergence shows the difference between the two sub-policies. We can observe that when $\alpha_2$ is significantly small or large (inclined to one sub-policy), the JS divergence is larger; while $\alpha_2$ is close to 0.5 (influenced by the two sub-policies equally), the JS divergence is small. Besides, we performed single sub-policy in rollout test. The agent with only executing $\pi_1$ can move flexibly without shooting any enemy, while the agent with only executing $\pi_2$ can kill the enemies but fall easily. Therefore, the framework has learned different meaningful sub-policies without specify the objectives of each sub-policy artificially.

In addition, we also observed that agent has learned a variety of combat policies. The agent tends to shoot at long distances when facing a single enemy. When facing multiple enemies at the same time, the agent are more inclined to close combat. On the one hand, it can avoid being attacked intensively by moving. On the other hand, it can get health point packets while fighting to supplement its own consumption, so as to ensure continuous combat.

## 6    CONCLUSION

In order to research on practical problems with multi-task combination, we implement a first-person shooting environment with random terrain, which is a typical representative of such problems. To deal with complex and subtle relationships between multiple sub-tasks, we propose a Meta Soft Hierarchical reinforcement learning agent, in which the high-level policy learns to combine the low-level sub-policies end-to-end through meta-gradients. Experiments show that the proposed framework outperforms state-of-art baselines and learns different meaningful sub-policies and combine them appropriately. We provide the open-sourced environment and code to encourage further research.

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
