# OpenReview forum: "Hierarchical Meta Reinforcement Learning for Multi-Task Environments"
_ICLR.cc/2021/Conference — Reject_

### Official Review · AnonReviewer2 · 2020-10-28
**The proposed hierarchical reinforcement learning method is quite straightforward and has been well studied in the literature**

**Rating:** 3
**Confidence:** 5

**Review:**

This paper considers a FPS game that can be decomposed into two sub-tasks, navigation and shooting. A hierarchical meta RL method is introduced and the updating rules for sub-policies and meta parameters are provided. Experiments focus on this specific environment and hence the hierarchical structure is also specified as a meta controller over two sub-policies defined for navigation and shooting explicitly.

The proposed hierarchical RL method is not novel and indeed a most straightforward way to control sub-policies with a meta controller. The number of sub-policies, each of which is specified to solve an explicit sub-task, is fixed given the environment and the rewarding scheme is also clear that each sub-task has its own reward. The final policy is simply presented as linear combination of sub-policies. Actually, similar studies has been well studied in the literature, like Feudal networks and MAML, which are even more general meta learning methods to automatically find sub-policies. These highly related approaches are ignored from the discussion and not considered in the experiments as a baseline.

Another question is about the considered global reward, which is also formulated as linear combination of the sub-tasks' reward using the meta parameters \alpha_i's. Under such a formulation, the rewarding scheme is dynamically varying as the training goes. One concern is that in IMPALA the estimated values in vtrace or advantage might suffer large variance since at early stages the meta parameters are almost from scratch. Another concern is that this rewarding scheme naturally restricts the meta controller to select sub-policy with high immediate R_i at a specific time step, and hence it seems the meta parameters are almost determined by the rewarding scheme.

Experimental parts lack comparison of many related works as mentioned above. Moreover, only one specific environment is studied with only two sub-tasks. It is hard to see the generality of the method when scaling to cases where a large number of sub-tasks exist.

---

### Official Review · AnonReviewer3 · 2020-10-28
**Several important missing details regarding environemnts and experiments, write-up needs to be improved.**

**Rating:** 3
**Confidence:** 4

**Review:**

This paper introduces a new first-person shooting environment consisting of two tasks: navigation and eliminating enemies via shooting. It describes a reinforcement learning architecture to that trains individual policies for each task and automatically balances between those policies by computing weights to mix the action distributions and the rewards.

Unfortunately, this paper leaves significant open questions about the environment and experiments. This makes it hard to judge the reported results. Chiefly, it is not clear what kind of reward should be maximized; the environment provides two rewards (one for each subtask), but Figure 4 and 5 compare algorithms for a single one-dimensional measure of performance. There are further questions around this: what kind of reward is used for the baselines (which are supposedly trained with a single reward)? What kind of reward does the value function for MeSH predict? The mixed reward as determined by the MetaNet? Then: The paper mentions that LSTM networks are used, but of what size? There are no details on complex baselines such as FuN, which are not trivial to implement or tune. Are the returns reported on a fixed set of levels? Figure 6 seems to imply that alpha weights sum up to one, but I couldn't find any description of this. Furthermore, details regarding the action space of the environment are missing.

The write-up would strongly benefit from some proof-reading to fix grammar and expression. Technical details such as the algorithm for procedural level generation or the observation space could be put in an Appendix.

---

### Official Review · AnonReviewer1 · 2020-10-28
**Interesting method, good performance, highly limited and ethically problematic evaluation domain**

**Rating:** 4
**Confidence:** 4

**Review:**

The paper proposes a meta-learning approach for hierarchical reinforcement learning, essentially meta-learning parametrised weights of a high-level controller. The method is tested on a single, new environment (a first person shooter) against a small set of baselines.

This review will focus on the method but I personally find that this application with particular emphasis on ‘killing’ (as terminology from the paper), is highly inappropriate and the method could easily be tested on a less problematic domain.

Overall, the paper is clearly written and the additional analysis on JS divergence and alphas in Section 5.3 provides an interesting perspective (which sadly turns into an analysis of semantic aspects of the game in terms of killing and how different amounts of enemies are handled).

A main shortcoming of the submission is the evaluation section. Even ethical concerns aside, additional baselines from the existing HRL literature would be beneficial as e.g. [2]. In addition, another baseline MLSH (Frans et al 2017, referred to quickly in the submission) would be a good baseline. Meta-learning the high-level while applying IMPALA for the low-level (proposed method) is highly related to MLSH which meta-learns the low-level while learning the high-level individually for each task.

One core argument of the paper surrounds the combination of multiple policies in a single time step vs ‘hard’ switching between policies across time-steps. The combination of objectives for every output is familiar in the multi-objective RL literature [1] as well as other methods in hierarchical RL [2], work that would provide stronger context for the investigation.

Finally, the evaluation focuses on a single task which has been introduced for this paper, strongly limiting the viability of the analysis. Further multi-task domains could be taken from the existing multi-objective/multi-task RL literature as well as via simple extensions of existing domains (imagine a cartpole environment with additional action penalty).

The final controversial point of this submission is the fact that the weights for combining rewards, which seem to also be used for the overall evaluation of the agent itself (though this remains ambiguous in the submission), are learned by the agent. This could on one hand lead to the agent finding a shortcut and ignoring some given objectives and on the other hand renders these weighted returns irrelevant for comparing different agents as the weights underlying the returns are part of the agent.

Overall, the paper proposes an interesting method but the analysis has ethical problems as well as only a single environment which was created purely for this submission and limited baselines. While the main ideas underlying the method remain interesting, all these problems should be addressed before considering publication.


[1] Vamplew, Peter, et al. "Empirical evaluation methods for multiobjective reinforcement learning algorithms." Machine learning 84.1-2 (2011): 51-80.
[2] Peng, Xue Bin, et al. "MCP: Learning composable hierarchical control with multiplicative compositional policies." Advances in Neural Information Processing Systems. 2019.

---

### Official Review · AnonReviewer4 · 2020-10-30
**Interesting idea but not enough clarity to really understand what is going on.**

**Rating:** 3
**Confidence:** 3

**Review:**

First of all I want to point something out I found quite bothersome:
The abstract states " A desirable agent should be capable of balancing between different sub-tasks: navigation to find enemies and shooting to kill them."  and the intro begins with "...it is an urgent need to use DRL methods to solve more complex decision-making problems."  I want to state that I strongly believe we should not be framing our research problems with these types of problems, nor trivializing concepts such as killing 'enemies'.  I'll try to be as unbiased in my scientific evaluation of this paper, but I would request that the language be toned down a bit, and ideally other types of tasks considered down the road.

Back to the review:
This paper presents a multi-task agent architecture with a final mixture component.  The authors show that this approach can work on a custom-built FPS game better than competing SoTA methods (both multi-task and mono-task agents.  Overall the method uses a bi-level optimization to find the optimal mixture of sub-policies as well as individually optimize each sub-policy.

Pros:
This seems like a relatively simple architecture and the empirical results are promising, and the analysis of alpha values correlation to sub-tasks is interesting and seems to indicate that the meta-controller does gain some insight into sub-task structure.

Cons:
There are many confusing points about the paper that made it hard to follow and that I would need clarified to argue for acceptance:

1. Doesn't min(clip(ρ) · A, ρ · A) = clip(ρ) A ?
2. Doesn't $\grad \alpha L train(ω + \epsilon , \alpha) − \grad \alpha Ltrain(ω + \epsilon, \alpha) = 0$. Perhaps you meant \grad w on the second term?
3. What are L_train and L_val?  I couldn't find a clear definition.
4.  Do the sub-agents receive the values of the respective R_i or is only $R = \sum_i \alpha_i R_i$ passed to the agent?
5.  I am not familiar with bi-level optimization, it could be worth talking a bit more about this instead of environmental architecture choices which are relatively irrelevant to the core of the paper.
6. Could you come up with  a task with more than 2 sub-tasks?  I find that 2 is likely a corner case and going beyond just two policies would make the method more convincing.  Trying out on a smaller more synthetic environment that would more easily allow task factorization would also allow for better empirical evaluation.
7. Although I am not super familiar with the FPS environments for RL, are there some already existant environments that have been independently benchmarked?  Any FPS env has a mixture of navigation and fighting, so I wonder what the value of proposing a new environment for this would be.

Conclusion:
I am not an expert of the multi-task literature but although the proposed idea seems to have merrits, the paper would need to be more clear on the points above for me to consider it clean enough for publication.  As it stands the approach is too opaque to really understand what is going on.

---

### Decision · Program_Chairs · 2021-01-07
**Final Decision**

**Decision:**

Reject

**Comment:**

The reviewers agreed that the paper presents interesting ideas but the presentation of the paper needs be improved. Also, the experiments and the related work section need be improved.